# Cognitive Radio with Machine Learning to Increase Spectral Efficiency in Indoor Applications on the 2.5 GHz Band

**DOI:** 10.3390/s23104914

**Published:** 2023-05-19

**Authors:** Marilson Duarte Soares, Diego Passos, Pedro Vladimir Gonzalez Castellanos

**Affiliations:** 1Instituto de Computação, Universidade Federal Fluminense, Niterói 24210-310, Brazil; 2ISEL, Instituto Superior, de Engenharia de Lisboa, 1959-007 Lisboa, Portugal; 3Departamento Engenharia de Telecomunicações, Universidade Federal Fluminense, Niterói 24210-240, Brazil

**Keywords:** opportunistic system, spectral efficiency, indoor applications, 2.5 GHz LTE, machine learning

## Abstract

Due to the propagation characteristics in the 2.5 GHz band, the signal is significantly degraded by building entry loss (BEL), making coverage in indoor environments in some cases non-existent. Signal degradation inside buildings is a challenge for planning engineers, but it can be seen as a spectrum usage opportunity for a cognitive radio communication system. This work presents a methodology based on statistical modeling of data collected by a spectrum analyzer and the application of machine learning (ML) to leverage the use of those opportunities by autonomous and decentralized cognitive radios (CRs), independent of any mobile operator or external database. The proposed design targets using as few narrowband spectrum sensors as possible in order to reduce the cost of the CRs and sensing time, as well as improving energy efficiency. Those characteristics make our design especially interesting for internet of things (IoT) applications or low-cost sensor networks that may use idle mobile spectrum with high reliability and good recall.

## 1. Introduction

The licensed spectrum is limited, making new licenses scarce and expensive. At the same time, the proliferation of internet-connected devices can make the industrial-scientific-and-medical (ISM) bands highly congested [1,2]. CR, dynamic spectrum access (DSA), and ML are considered a possible solution, as they increase the spectral efficiency of currently available frequencies using “spectral windows” [3,4,5]. These windows are spatial and/or temporal regions in which the spectrum is not used by the primary users (PUs). This allows secondary users (SUs) to use the spectrum vacancy to transmit. In other words, SUs dynamically accesses the licensed spectrum when (or where) it is temporally available [2,3,4,5,6,7,8,9].

CR can be considered the solution to the spectrum wastage due to low usage of the licensed radio spectrum, because this technology works on the principle of adapting the operating parameters of the radio according to the conditions of the environment. Thereby, it facilitates smart, efficient, and reliable spectrum usage through identifying and exploiting a large amount of unused or underused spectrum bands [7,8].

At the same time, the IoT continues to gain importance and popularity due to the ever-growing number of low-price and low-power smart devices, like sensors, actuators, and radio frequency ID (RFID) tags, that can connect to the internet anytime, anywhere and for anything. It is expected that, by 2025, the number of internet-connected devices will be 10 times larger than the human population [2,6]. IoT devices are, therefore, anticipated to demand a huge amount of wireless spectrum in the near future, which contrasts with the current issue of spectral scarcity [2,6,8].

Many of the IoT devices will operate in indoor environments (e.g., devices for home automation, sensors deployed in industrial plants, healthcare, smart cities, and smart homes) [8]. Indoor environments are particularly prone to spectral windows, as coverage by PUs is strongly influenced by the constructive characteristics of buildings. Additionally, in the particular case of mobile networks, national regulatory agencies often do not impose indoor coverage requirements on the incumbents [10,11].

When requirements exist, the main parameter used to verify the system coverage is the received signal level. In the case of the long term evolution (LTE) standard for mobile broadband communications, the reference signal received power (RSRP) is used for this purpose. The RSRP is defined as the average of the energy contributions from all resource blocks (RBs) within the incumbent’s bandwidth. An RB is the smallest unit of band resource that can be allocated to a user. In the LTE technology, it corresponds to a bandwidth of 180 kHz—incumbents with a total bandwidth of 10 MHz have 50 RBs, while those with 20 MHz bandwidth have 100 RBs [12,13,14]. Anatel—the Brazilian regulatory agency—for example, adopts RSRP ≥ −110 dBm as a criterion for outdoor coverage [10,11]. More precisely, an RSRP above or equal to −110 dBm must be present for more than 90% of the time for an outdoor area to be considered covered. However, studies on LTE networks show that RSRP values ≤ −95 dBm generate unacceptable service quality levels, resulting in call drops for −108 dBm ≤ RSRP ≤ −100 dBm, in particular, throughput is almost zero [15,16].

In this paper, we seek to design an autonomous and decentralized CR, operating on the 2.5 GHz band—used by 4G systems worldwide with billions of subscribers—in the indoor environment, which often has poor signal coverage. Towards that goal, we address the following open questions:How to decide when to access the entire bandwidth of the incumbent in a time shorter than the LTE’s frame duration?Once detected as free, for how long can the CR use the bandwidth?How to measure in loco the incumbent’s occupancy rate, to improve the CR’s decision?What machine learning (ML) classifier algorithms can perform well in detecting transmission opportunities on LTE bandwidths based on information provided by a small number of energy detectors?Is there a minimum number of energy detectors that allows the detection of transmission opportunities with satisfactory performance?

To that end, we collected and analyzed—through statistical and ML tools—data from measurements of three incumbents with different bandwidths and traffic loads, using a spectrum analyzer in the indoor environment.

Our results indicate that our proposed CR with ML model can predict access opportunities with at least 63% recall and more than 91% reliability, with spectral windows ranging from 5 ms up to 85 ms, in occupancy rates up to 65%. In addition, it has over 80% efficiency in transmitting 5 ms packets for all of the evaluated incumbents. We argue that these efficiency levels are sufficient to enable IoT or similar applications to share the same band with mobile communication networks, improving spectral efficiency in indoor environments. Motivated by that, we also propose an architecture for a CR with five or ten energy detectors based on these models and considering physical limitations.

The contributions of this work are summarized as:(1)Present a CR solution that can use the licensed 2.5 GHz spectrum—independent of any mobile operator or external database—with over 90% reliability and over 60% recall, with sensing only 10% of the RBs in the LTE incumbent’s bandwidth.(2)Present statistical modeling that can use real data dynamically collected—with a reduced number of sensors—in the environment to increase spectral efficiency, enabling the insertion of new services in the 2.5 GHz band, such as IoT.(3)Prove that it is possible to use a licensed spectrum (mobile area) that has a low occupancy rate through statistical modeling and ML, obtaining high reliability and good recall.(4)Create a concept that can also be used in areas not fully covered by incumbents, such as rural areas. Thus, enabling the evolution of wireless services in agriculture and small cities.

The rest of this paper is organized as follows. Section 2 gives an overview of related works on using a CR with ML applications, spectrum sensing, use of the licensed spectrum, and IoT. Section 3 describes the spectrum measurements and data acquisition. Section 4 presents the study of the idle periods in LTE bandwidth. Section 5 presents the machine learning models for CR access on LTE bandwidth. Section 6 presents an autonomous and decentralized CR architecture proposal, using ML models with the minimum number of energy detectors; then, its performance is evaluated and compared with the technical literature in Section 7. Finally, in the last section, we conclude the paper.

## 2. Related Works

By now, CR is a well-established concept, with abundant literature available. For the past several years, it has been possible to find examples of works that attempt to address how a CR can operate opportunistically in channels used by different PUs, where spectrum sensing is an essential component. CR, DSA, and ML algorithms are powerful techniques for designing a promising spectrum sensing model to improve IoT on the licensed spectrum [2,3,6,9,17]. For example, in [5] the authors used the licensed spectrum with the supervised ML algorithms support vector machine (SVM), k-nearest neighbor (kNN), and decision tree (DT) to detect the existence of primary users (PUs) over the television band. Moreover, principal component analysis (PCA) is incorporated to speed up the learning of the classifiers. In [7], the authors used television white spaces (TVWS)-based cognitive radio systems to develop an accurate probabilistic model for improving spectrum efficiency by facilitating opportunistic usage of the television broadcasting spectrum by secondary users without interfering with primary users. In [8], the authors achieved higher data rates and minimized end-to-end routing delays in CR-enabled IoT communication in order to maximize throughput. They proposed a reinforcement learning (RL)-based routing approach in the cognitive radio network (CRN)-based IoT environment.

Saving energy is important in all networks, therefore, in [18], transmission of SUs is handled with a deep deterministic policy gradient (DDPG), where ambient backscatter communications and radio frequency (RF)-powered cognitive radio networks address the concerns of energy and spectrum scarcities from different perspectives, and the integration of them has potential benefits for throughput. The other important solution was created in [19]: when the licensed spectrum is occupied, SUs harvest energy from the primary RF signal; when the spectrum is available, SUs transmit data by using the harvested energy.

In [20], the authors propose a cognitive engine design that enables a radio to find transmission opportunities in the non-contiguous wideband spectrum to avoid interference with CR, using multi-task transfer deep reinforcement learning that can be applied in the licensed spectrum to improve quality-of-service (QoS). The spectrum is partitioned into sub-bands, each made of a number of narrowband channels. A multi-task deep Q-network (DQN) is utilized to solve the underlying problem, where communication over each sub-band represents a single task.

In [21], the authors exploited an overlay ad hoc secondary network operating at a global system for mobile communication (GSM) bands. The CRs would sense the medium and use blocks of 200 kHz for slots of 0.57 ms. In [22], the authors present a proposal of a CR for IoT operating at the 3G bands, using spectrum occupancy prediction via long short-term memory (LSTM). This enables the use of 200 kHz channels within the 1835–1848 MHz range, with success rates above 70%. However, it does not predict for how long the channel will remain idle, and 70 sensors are needed so that a CR can cover the entire studied frequency range.

In [23], the authors propose a distributed Q-learning approach based on DSA as a viable and easily implementable solution for facilitating secondary LTE spectrum sharing in high-capacity dense cognitive cellular systems. The proposal aims at increasing the use of LTE frequencies but keeping all the control of the spectrum with the incumbent. Similarly, ref. [24] explores the idea of an ML-based spectrum sharing system in which LTE is used both on a primary network and on a secondary network. Thus, LTE cell phones can detect opportunities and predict the time when RBs are idle in the secondary network, to occupy them with the use of the multilayer perceptron algorithm and LTE coordination. Ref. [25] looks for the convergence of wireless sensor networks (WSN) and cognitive cellular networks using frequency reuse of LTE. The authors introduce a CR-based algorithm for small nodeB (SeNB) to opportunistically offload devices over macrocell.

A DSA framework for an LTE-Advanced network is presented in [26,27]. The CR listens to one RB for 6 ms and decides whether or not to use it for 40 ms. However, only that particular RB can be used, and not the entire bandwidth of the incumbent. The authors in [26] highlighted the existing blocks in LTE-A networks that can be used for supporting CR concepts. According to them, DSA can be achieved just by introducing a few additional blocks to the existing network. The authors in [27] utilize the framework presented in [26] and introduce opportunistic spectrum access in an LTE-A network. Their work illustrates the adoption of a geolocation database within the LTE-A network that gathers cooperative information from the CR users about their surrounding environment.

Another article about DSA [28] presents channel prediction with the multilayer perceptron neural network (MP) and support vector machine (SVM) algorithms, obtaining accuracies above 85% and up to 92.9% for channels with occupancy rates of 50% up to 80%. The authors also present models for estimating the average idle time of the channel. The work is not concerned with analyzing any specific channel or technology and uses three traffic models (Poisson, interrupted Poisson, and self-similar) for PUs. The article only tests synthetic traffic and does not take into account the nature of LTE channels composed of several, possibly independent, RBs. While the proposal could be adapted for such a scenario by treating each RB as an independent channel, this would require 50 or 100 energy detectors to monitor the full bandwidth of the incumbent, while also restricting the used bandwidth to that of a single RB.

Like [28], our work seeks a decentralized and autonomous ML-based CR solution that does not depend on coordination with the incumbent’s network or the existence of a geographical database. Unlike [28], however, our work considers the specificities of the LTE channels, particularly their division into RBs and their frame duration, while also seeking to use the full bandwidth of the incumbents. Moreover, both our proposal and evaluation are based on real LTE bandwidth data, instead of assumed traffic patterns. We also contributed by reducing the number of energy detectors per bandwidth, because in the analysis of each 10 MHz or 20 MHz bandwidth, a methodology was used that allows sensing part of the bandwidth without a loss in reliability, reducing the sensing time.

## 3. Spectrum Measurements and Data Acquisition

A measurement campaign was carried out in the city of Niteroi in the state of Rio de Janeiro, Brazil, with the objective of survey spectrum occupation. The measurements were carried out inside the engineering building of the University Federal Fluminense (UFF), the 2.5 GHz band follows the 3GPP standard and the frequency division duplex (FDD) is used, as can be seen in Figure 1, with uplink and downlink for each incumbent (Incumbent_1 was not in operation at that moment). The period in which the data were collected was approximately 24 h, according to the methodology presented in [4,29,30,31,32,33,34]. There are also portions dedicated to the time division duplex (TDD) that were not in operation in Rio de Janeiro. As the figure shows, the incumbents can be classified according to how much bandwidth they have allocated. In particular, the figure shows incumbents with 10 MHz and 20 MHz of bandwidth. Regardless of bandwidth, however, the FDD LTE system uses radio frames with a duration of 10 ms [12,13,31]. It is important to mention that not all of the band is used by the incumbent’s signal, there is a portion of the spectrum that is left free (guard band) to avoid interference between adjacent channels [12,13,14,31].

Our measurements center around two main values of interest: the received signal strength indicator (RSSI) and the RSRP. The RSSI is a parameter that provides information about the total power received over broadband. In practice, it is the channel power over the incumbent’s bandwidth. In other words, it is the sum of the contributions of the energies of all of the RBs [12,13,14,29,30,31,32].

On the other hand, the RSRP is the average channel power considering each of the incumbent’s RBs, as described in Equation (1), where *n* = 50 or 100 for 10 or 20 MHz, respectively, depending on the number of RBs on the bandwidth [12,13,14,29,30,31].
(1)$$RSRP\left(dBm\right)=RSSI\left(dBm\right)-10log\left(n\right).$$

### 3.1. Measurements Setup and Occupancy Decision

The measurements were carried out with the Anritsu MS2692A spectrum analyzer with the internal preamplifier in the off state, an omni-directional antenna AH-8000 wideband 100 MHz to 3300 MHz (3 dBi), a low-noise amplifier (LNA) with 14 dB, two coaxial cables (0.3 dB/m) of 50 cm each, which connect the antenna to the LNA and the LNA to the analyzer, and a laptop, with MATLAB, powered by a no-break. The spectrum analyzer cyclically swept the band in the range from 2595 MHz to 2695 MHz, using 1001 points—i.e., a 100 kHz frequency bin—guaranteeing at least one reading per RB, in agreement with the channel arrangement of the 3rd Generation Partnership Project (3GPP) [12,14]. The revisit time was 3.14 s, and the measurements targeted at least 95% confidence levels at various occupancy percentages [4,32,33,34,35]. Hereinafter, we refer to each of these sweeps as a trace. Figure 2 illustrates one such trace, where it is important to observe the amplitude in the energy variations in the RBs that will feed the datasets used in the ML. The 24 h measurement period resulted in a total of 27,504 traces (1146 traces per hour).

We restricted the analysis to the downlink channel of each operator channel. We argue that a lack of signal at the downlink implies that the uplink is idle as well, because without a “useful” signal from the incumbent on the downlink, there is no reason for the subscriber to use the uplink—it is important to note that handset power is around 1 watt and the base transceiver station’s (BTS) power is 50 watts. Because of this, we believe that future studies should further evaluate this hypothesis. 

In the analysis, the data of each incumbent is analyzed independently. In the post-processing step, the levels of signal received through the collected traces are processed and transformed into RSRP decision thresholds to assess whether their bandwidth is busy or idle. That is, if the RSRP is above or equal to the threshold value, the bandwidth is considered busy, otherwise it is considered idle. In order to evaluate the effect of this threshold in our proposal, we considered threshold values within the range −110 dBm ≤ RSRP ≤ −91.5 dBm, with steps of 0.5 dB, depending on the incumbent’s power signal. The distance between the average value of the noise floor and the decision threshold improves the precision in decision making, avoiding a false alarm error. The choice of setting the resolution bandwidth (RBW) at 10 kHz followed the concept of using a noise floor of around 3 dB below the lowest decision threshold, or RSRP = −110 dBm [29,32,35].

### 3.2. Making a Database with Trace Information

We started by splitting each trace in order to separate the information for the bandwidth of each different incumbent (e.g., the downlink of Incumbent_2 ranges from 2630 MHz to 2650 MHz, the downlink of Incumbent_3 ranges from 2650 MHz to 2660 MHz, and the downlink of Incumbent_4 ranges from 2660 MHz to 2670 MHz). For each trace and each incumbent, we then applied the RSRP decision threshold, to assess whether its bandwidth was busy or idle at that moment. Based on that, vectors with spectrum opportunity information were generated, as illustrated in Figure 3. A vector represents the evaluation of the occupancy of an incumbent for a certain RSRP decision threshold. Each position of a vector represents the state of the bandwidth—busy (1) or idle (0)—for a particular trace.

Due to the signal occupancy variability in time, we also computed the average occupancy of each incumbent’s bandwidth during certain periods of the measurements (e.g., during the full 24 h or for each 1 h). For that, we also resorted to the RSRP criterion. However, since LTE uses 10 ms frames and our scan time is larger than 5 ms, to achieve a certain confidence level when measuring the occupancy of a bandwidth, a given number of traces are required. There is a linear relationship between the occupancy rate and the number of traces required. The lower the occupancy, the more traces are needed to achieve the desired confidence level. The relationship between the number of traces required for a certain confidence level is detailed in recommendation ITU-R SM.182.4, recommendation ITU-R SM.1880-2, and recommendation CEPT/ERC 01-10 E, and reproduced in Table 1 for reference purposes (the table assumes a four-second revisit time) [4,33,34]. For example, with the 27,504 traces obtained during the 24 h measurement period, as each trace can be considered as an independent sample, any vector with at least a 2% occupancy rate can be considered reliable, as shown in Table 1 [4,29,30,31,32,33,34,35]. Similarly, for periods of one hour, the 1146 samples allow us to reliably estimate occupancy levels of at least 35%.

In this work, the incumbent’s occupancy rate is the percentage of time that its RSRP is above or equal to the RSRP decision threshold, where, by Equation (2), *n* = 27,504 for 24 h and 1146 for 1 h.
(2)$$Occupancy\;Rate=\frac{\sum_{i=0}^{n}IF\left({RSRP}_{i}\geq decision\;threshold\right)}{n}.$$

Note that the RSRP threshold directly affects whether the bandwidth is considered to be occupied by the PUs. Therefore, for each incumbent, several different spectrum opportunity vectors were generated using different decision thresholds, resulting in different occupancy rates. Figure 4 shows how the RSRP decision threshold (variation with 0.5 dB steps) generated each incumbent’s real average occupancy rates during the full 24 h period of the measurements.

### 3.3. Occupancy Variation during the Measurement Period

As can be observed in Figure 5, the occupancy rate varied for each incumbent during the measurement period—each point refers to the average occupancy during a one-hour period considering an RSRP decision threshold of −110 dBm. Incumbent_2, in particular, shows a large variability throughout the day, but all incumbents show some variability as well. We argue that these variations enrich the analysis presented in this paper, as they allow us to evaluate our proposal under very different circumstances (both in terms of occupancy and bandwidth). Moreover, Figure 5 also shows that Incumbent_5 presented very low occupancy levels throughout the day (0.71%, considering the 24 h period), meaning that we cannot rely on its occupancy figures given the number of traces that we have available. Thus, data from Incumbent_5 will not be considered in the remainder of this paper [29,30,31,32,33,34,35].

## 4. Study of the Idle Periods in LTE Bandwidth

After identifying the spectral opportunities, it is necessary to estimate for how long that bandwidth will remain idle. To help answer that question, a statistical analysis of the collected data was performed. For each incumbent, histograms of the duration of the idle moments were generated. As the LTE radio frame is 10 ms, we considered that each incumbent’s idle sample corresponds to 10 ms. Thus, in the histograms presented in this section, the idle periods are always multiples of 10 ms. The graph in Figure 6 was generated within 24 h of Incumbent_3’s samples, with a decision threshold RSRP = –102 dBm.

The histogram in Figure 6 shows an exponential downward trend. This behavior is consistent with an assumption that idle LTE frames occur independently, which would result in an exponential distribution of those idle period durations [28].

Given that assumption, one can choose a period, TMAX, for the maximum time the CR should use the bandwidth once it becomes idle, so that there is a relatively small probability that a PU will use the bandwidth before TMAX. To do that, it should be noted that the probability that the bandwidth will remain idle for at least TMAX is given by [28]: (3)$$ProbSuccess=\int_{TMAX}^{\infty }\mu e^{-\mu t}dt.$$

In this equation, 1/μ is the average idle period duration [28]. If a desirable reliability level is chosen, hereinafter called simply *ProbSuccess*, and if we consider the fact that an LTE frame is 10 ms long, TMAX could be solved by (3) as:(4)$$TMAX\left(s\right)=\frac{-ln\left(ProbSuccess\right)}{\mu }+0.01.$$

In practice, the μ parameter of the distribution can be estimated in loco by computing the sample mean of the duration of the idle periods as monitored by the CR. Further, note that because LTE uses 10 ms frames, the values output by Equation (4) can be truncated to the nearest multiple of 10 ms.

## 5. Machine Learning Models for CR Access on LTE Bandwidth

One of the basic challenges for a CR is to identify the secondary access opportunities. For this purpose, in this paper, we proposed the use of ML to classify the incumbent’s bandwidth into busy or idle states, based on energy level readings of the incumbent’s RBs. While other works have followed this idea before, we note that, given the typical bandwidth of an incumbent, the number of RBs is large—e.g., 50 for a 10 MHz bandwidth. Thus, monitoring all RBs might be technically or economically unfeasible. Instead, we also investigated how to reduce the number of monitored RBs required to achieve good classification performance with a reduced number of energy detectors. It is also important to note, that the sampling of some RBs or portions of the channel spectrum is not fully affected by frequency selective fading, as the spectrum portions selected for sampling are in different parts of the channel.

The first datasets used for training and testing the ML models are composed of the data described in Section 3. We generated different datasets for different incumbents and for different values of the RSRP threshold. Each instance of a dataset corresponds to the state of the incumbent’s bandwidth during one of the 27,504 traces we collected (see Figure 3). More specifically, the trace is composed of 50 or 100 features for incumbents with 10 MHz or 20 MHz bandwidth, respectively, one for each RB. Each feature is the received signal level measured for the corresponding RB in dBm. The class of each instance is the information of whether the incumbent’s bandwidth was busy (1) according to the RSRP decision threshold (considering all RBs) or idle (0). Figure 7 illustrates part of one of the assembled datasets. In the rest of this paper, we will refer to the possible outcomes of a bandwidth classification by an ML model as True Idle or TI (the bandwidth was idle and was classified as idle), False Idle or FI (the bandwidth was busy, but was classified as idle), True Busy or TB (the bandwidth was busy and was classified as busy), and False Busy or FB (the bandwidth was idle, but was classified as busy) [5,36,37].

### 5.1. Metrics for Evaluating Machine Learning Models with All RBs

All of the experiments reported here were performed in Python with Sklearn and standard parameters, where each tested dataset used 30% of the data for training. After checking the technical literature about CR for spectrum sensing and spectrum prediction [5,17,28,37,38,39,40,41,42], we obtained datasets with different occupancy rates by changing the RSRP decision threshold with 0.5 dB steps and the information of all of the 50 or 100 RBs. Thus, we evaluated the following classifiers: naïve Bayes (NB), random forest (RF)*,* multilayer perceptron neural network (MP), and support vector machine (SVM) algorithms. These classifiers were selected to cover a wide range of algorithm families [5,17,37,38,39,40,41,42].

Our study does not seek to limit the search for better classifiers, therefore, the methodology developed here can be applied to other classifiers, including deep learning, thus making it possible to evaluate classifiers with higher returns in terms of spectrum efficiency in LTE channels. 

Important ML performance metrics are applied to the datasets, among them, considering the scenario where the CR looks for opportunities in a licensed LTE band, the number of False Idles is a very important performance metric, because it is necessary to disturb the PUs as little as possible. Otherwise, it increases the number of times that the incumbent´s bandwidth is busy and the ML algorithm informs the CR that the band is free. Equation (5) presents the reliability of the prediction in all traces analyzed. More specifically, we define the reliability of the classifier as:(5)$$Reliability=\frac{ALLtraces-FI}{ALLtraces},$$
where ALLtraces denotes the total number of traces classified.

Figure 8 presents the reliability results of channel classification for Incumbent_2, Incumbent_3, and Incumbent_4, with twenty-four hours of collected data. Each plot shows separate lines for each classifier, while the horizontal axis shows the real occupancy rate for a given incumbent.

Another important performance metric in this context is recall, which is defined as [5,36]:(6)$$Recall=\frac{TI}{TI+FB},$$
where TI and FB mean True Idle and False Busy. In simple terms, recall represents the success rate when the classifier states that the bandwidth is idle and the incumbent’s bandwidth is idle. Figure 9 presents the recall results of bandwidth classification for each incumbent, with twenty-four hours of collected data, with different occupancy rates obtained by changing the RSRP decision threshold with 0.5 dB steps and the information of all of the 50 or 100 RBs.

Figure 8 and Figure 9 show that the results are rather consistent regardless of the bandwidth or data traffic type in each incumbent. However, these figures show that the performance in both metrics is dependent on the classification algorithm and that MP is too unstable to use on these datasets. More interestingly, there is not a single algorithm that outperforms the others in all or most cases. Instead, the choice of the best algorithm seems to be dependent on the average bandwidth occupancy rate. For instance, naïve Bayes is the best choice in terms of reliability for lower real occupancy rates (i.e., lower than 30%), but as the bandwidth becomes busier, the SVM and RF algorithms become the better choice. Thus, a good alternative would be for the CR to adapt to the environment by taking into account the incumbents’ occupancy and dynamically choosing the best classifier accordingly.

Further, note that the occupancy rate can be easily estimated in loco based on the output of an ML classifier by simply computing the ratio between the numbers of the traces classified as busy to the total number of traces classified.

In order to evaluate the accuracy of this estimate, we define the error, ∆, as the difference between the real occupancy rate of an incumbent and the estimated occupancy rate with ML:(7)Δ=REALocup−MLocup.

Figure 10 shows the ∆ obtained with each of the evaluated ML algorithms for the incumbents as a function of the real occupancy rate, with 50 or 100 RBs. Overall, the SVM and RF algorithms have a higher accuracy (i.e., lower ∆), typically within 5% of the real value—the two classifiers being less sensitive to changes in occupancy rates and operator traffic. These graphs confirm the research presented in [17], in which it was reported that spectrum sensing is a task well-performed by algorithms such as kNN, Q-learning, RF, and SVM.

Accuracy is a metric that generally describes how the model performs across all classes (busy or idle in our case). It is useful when all classes are of equal importance. It is calculated as the ratio between the number of correct predictions and the total number of predictions [5,36,37].
(8)$$Accuracy=\frac{TI+TB}{ALLtraces}.$$

Figure 11 presents the accuracy results of the bandwidth classification for Incumbent_2, Incumbent_3, and Incumbent_4 with twenty-four hours of collected data, with different occupancy rates obtained by changing the RSRP decision threshold with 0.5 dB steps and the information of all of the 50 or 100 RBs.

In addition, it is observed in Figure 10 and Figure 11 that the SVM and RF algorithms have good accuracy in all ranges of real occupancy rates, enabling the use of these algorithms to adapt the CR to the environment, guaranteeing the reconstruction of the original incumbent’s signal (idle or busy) with good fidelity. Taking into account the execution time obtained during the tests with the datasets, random forest performed faster, generating a lighter model for execution in an embedded system. Therefore, the paper from this point on will work on detailing our CR solution considering naïve Bayes and random forest. It is important to note that other combinations with NB, SVM, and RF may be tested in the future, taking into account the efficiency and application of a cognitive solution to a specific environment, as well as other classifiers not evaluated here, including deep learning. 

### 5.2. Evaluating Machine Learning Models to Reduce the Number of Monitored RBs Required

While the results of Section 5.1 are encouraging, they are still based on datasets that contain information regarding all of the RBs of the incumbent—50 or 100, depending on the bandwidth. In order to make our CR solution more feasible—both technically and economically—it would be beneficial if we could consider only a subset of the RBs, such that the CR could rely on a reduced number of energy detectors to evaluate the full bandwidth of the incumbent. To this end, we now evaluate a feature selection based on the GainRatio filter, which ranks the RBs according to the amount of information they convey regarding the state of the incumbent’s bandwidth. 

#### 5.2.1. Applying the GainRatio Filter to Generate Datasets with 10% of the Incumbents’ RBs

We started by applying the GainRatio filter—on datasets with 50 or 100 RBs—to all incumbents’ 24 h samples for each decision threshold (−110 dBm to −102 dBm, with 1 dB steps), and grouped the top 15 best-ranked samples at each level (or each dataset). Then, we observed which were the 5 that appeared most frequently in the 135 tabulated samples, and in the best positions. Figure 12 shows the process applied in choosing the five RBs for Incumbent_3.

With 5 or 10 best RBs of each incumbent, we must cut the other columns with information from the other RBs in the datasets that are generated for each RSRP decision threshold (represented in Figure 7) and keep the class column, that contains the information of the state of the incumbents’ bandwidth (idle or busy).

#### 5.2.2. Results Achieved with 10% of the Incumbents’ RBs

Figure 13 presents the accuracy values in the three incumbents, applying the datasets with 10% of the RBs—based on the 24 h of collected data—with the NB and RF algorithms. The results also show high accuracy values across the occupancy range, keeping good perspectives for the development of our CR solution.

Figure 14 also reinforces the use of RF for in loco detection of the occupancy rate, as it presents the best accuracy when compared to SVM in datasets with 10% of the RBs of each incumbent—keeping ∆ typically within 5% of the real value—mainly in real occupancy rates below 30%, where the precision of the in loco evaluation must be high. These graphs confirm the research presented in [17], in which it was reported that spectrum sensing is a task well-performed by algorithms such as kNN, Q-learning, RF, and SVM.

## 6. An Autonomous and Decentralized CR Architecture

Therefore, based on the results reported in Section 5, and the possibility of obtaining good results by sensing a few RBs in the incumbent’s bandwidth, an architecture proposal was developed for an autonomous and decentralized CR capable of operating in the 2.5 GHz band alongside LTE. Figure 15 summarizes the design for operating with energy detectors for a subset with different RBs (5 or 10), which constantly provide samples to embedded ML models that estimate incumbent occupancy rates and detect transmission opportunities. 

Each energy detector outputs the state of its respective RB in dBm. This information of subsets’ RBs are fed into blocks 2 and 3, which execute previously trained ML models based on the NB and RF algorithms, respectively. Both blocks run in parallel and output the classification of the state of the incumbent’s bandwidth—again, either 1 (busy) or 0 (idle).

These classifications are then fed into block 4, which serves multiple purposes. Firstly, it uses the output of the classification by block 3—the random forest—to estimate the current occupancy rate. Based on that occupancy rate, it implements the mathematical model proposed in Section 4, which, in turn, estimates the maximum channel access time, TMAX, based also on the stipulated reliability level (ProbSuccess)—received as an input through In_Success—allowing for a dynamically reconfigurable system. If the estimated occupancy rate is below 30%, then block 4 uses the output of block 2—naïve Bayes—to detect when the bandwidth is idle and, therefore, when the CR may begin transmitting. Otherwise, this idle information is taken from block 3—random forest—as this algorithm performs better under higher occupancy rates in our simulations. This bandwidth idle information is then combined with the estimated TMAX to output through Out_1 whether the CR can transmit at any given moment. 

According to the ITU-R SM.1880-2, 48,000 samples are required to perceive occupancy rates of at least 1% in a statistically significant manner. Because of the inter-sample time of 5 ms (used in block 1), a total sampling time of 4 min is used in block 4 [4]. We argue these values achieve a good compromise between tracking bandwidth changes and maintaining the technical feasibility of the energy detectors [43].

Another consequence of the inter-sample time of 5 ms is that block 4 must subtract 5 ms from the estimated TMAX. This is because, once the bandwidth becomes idle, 5 ms of the expected idle duration will be spent by the energy detectors.

## 7. Evaluation and Comparing the Performance of CR Architecture Proposal with the Technical Literature

In order to evaluate the proposed CR architecture, we developed a simulator in Python with Sklearn and standard parameters that implements our proposal. It receives an input line by line of a dataset of our measurements (27,504 lines), and outputs a number of performance metrics. The datasets are inserted in the simulator sequentially, that is, the second dataset of the queue is only inserted after the entire training process and output of the results of the performance metrics of the first one is completed. Each dataset used 30% of the data to perform the training of the NB (block 2) and RF (block 3) ML models. 

The simulator traverses the set of lines and treats each line—which represents 10 ms of the incumbent signal—as a reading by the energy detectors of each RB, that are grouped into subsets of 5 or 10. It then processes those readings according to the CR architecture in order to decide if the bandwidth is idle and for how long it will remain in that state. Based on that, the simulator further decides whether or not to transmit packets. Each packet is assumed to have a fixed duration of 5 ms.

At the end of the simulation, we also count the number of successfully transmitted packets (Packets_Tx) and the number of collisions (Packets_Col). A collision is defined as a transmission that took place because the proposed CR architecture predicted that the bandwidth would be idle when it was actually busy. From those two values, we further define the efficiency of the CR as:(9)$${Efficiency}_{5ms}=\frac{Packets\_Tx}{Packets\_Col}.$$

Despite not being the focus of the paper, possible transmission rates of the proposed CR architecture can be obtained through information on successfully transmitted packets (Packets_Tx), where one can consider the modulation schemes and coding systems currently most used in radio frequency equipment.

### 7.1. Evaluating the Mathematical Operations of the Simulator’s Block 4 in All Incumbents

Figure 16 presents the values obtained by the simulator for the spectral windows from the datasets of 24 h of the three incumbents at different real occupancy rates—i.e., considering different datasets created for different RSRP decision thresholds—and for different reliability levels (ProbSuccess). It is important to note, that above 15% real occupancy rate the spectral windows for CR access to the incumbents’ bandwidth is always 5 ms and the autonomous CR can dynamically observe that it does not need to perform the mathematical operations of block 4, saving internal memory and central process unit (CPU) time.

### 7.2. Evaluating Output Performance Metrics of Simulator for Each Incumbent

The evaluation used datasets for each incumbent of five or ten RBs—representing the readings of five or ten energy detectors—with 27,504 lines (traces) and the ProbSuccess adjustment with 99%, 95%, and 90%. Each incumbent was simulated independently, that is, at different times, where each incumbent used only its datasets, both for training and for output performance metrics.

The results clearly show the switching between the RF and the NB algorithms around a 30% occupancy rate—keeping ∆ typically within 5%—with a drop in recall, however, the reliability values continue to grow, which for our study is the most important, as we want to minimize the impact on the incumbents. They also show that there is a high reduction in packet transmission efficiency at real occupancy rates greater than 65%.

#### 7.2.1. Evaluating Performance Metrics with Ten Energy Detectors in Incumbent_2 (20 MHz Bandwidth)

Figure 17 indicates that the proposed CR architecture can predict access opportunities with at least 87.4% recall, and more than 91.1% reliability, obtaining high values in the entire occupancy range of Incumbent_2, enabling the SUs’ use of the bandwidth with reduced impact on PUs.

Figure 18 presents the number of successfully transmitted packets and packet transmission efficiency versus real occupancy rates. The proposed CR architecture had a high score, with at least 85.1% efficiency and high transmission rates of 5 ms packets.

#### 7.2.2. Evaluating Performance Metrics with Five Energy Detectors in Incumbent_3 (10 MHz Bandwidth)

Again, the proposed CR architecture simulator achieved a high score, as can be seen in Figure 19. The results show that CR can predict access opportunities with at least 68.5% recall and more than 91.5% reliability, that is, keeping the reduced impact on PUs as seen for the results of Incumbent_2.

Figure 20 maintains the high results that had already been achieved in Incumbent_2, proving that the proposed CR architecture performs well in a 10 MHz channel in a different range of real occupancy rates—monitoring 10% of RBs guaranteed similar performance—it is worth noting that lower occupancy rates, or equal to 65%, obtained at least 81.4% efficiency, keeping high transmission rates of packets of 5 ms.

#### 7.2.3. Evaluating Performance Metrics with Five Energy Detectors in Incumbent_4 (10 MHz Bandwidth)

Despite having a 10 MHz bandwidth, like Incumbent_3, and also using five energy detectors, Incumbent_4 has a wider range in occupancy rate and different traffic, enriching and confirming the results found in other incumbents. Figure 21 maintains more than 93.5% reliability; a high reduction in impacts on PUs ensures that the proposed CR architecture can predict access opportunities with at least 63.7% recall at real occupancy rates less than or equal to 65%.

Figure 22 reinforces the rapid decay in efficiency in the transmission of 5 ms packets in bandwidth with occupancy rates above 65%. Therefore, with rates up to 65% it obtained at least 83.6% efficiency, making this value an interesting limit for the proposed CR architecture.

### 7.3. Comparing the Proposed CR Architecture’s Performance with the Technical Literature

In this section, the results of the proposed CR architecture are compared with the technical literature model DSA [28], which presents bandwidth prediction with the MP and SVM algorithms (these will be referred to as BaselineMP and BaselineSVM in this section). To accomplish this task, a new simulator was developed in Python with Sklearn, where each tested dataset used 30% of the data to perform the training of the BaselineMP or BaselineSVM models. For the implementation of the BaselineMP algorithm, we used 4 inputs in the input layer, two hidden layers consisting of 15 and 20 neurons, and one neuron in the output layer. We used Adam and Relu as the solver and activation functions, respectively, for the hidden layer and the output layer neurons. The learning rate and maximum iterations were taken as 0.001 and 200, respectively.

For training and testing purposes, the “class” column of the datasets generated with the spectrum analyzer was used. Real LTE bandwidth data, where a sliding window with four inputs (idle or busy) was applied and the fifth is the result that must be achieved (idle or busy), was applied either to BaselineMP or BaselineSVM. Thus, it was possible to use the models described in the technical literature in the LTE bandwidth (10 MHz or 20 MHz) with only one energy detector. Therefore, the simulator senses the bandwidth four times and predicts if it will be busy or idle.

The reliability and efficiency to transmit the packets are very important metrics when using a CR which is trying to fill and to use the idle opportunities of an incumbent that has a licensed spectrum. Thus, in the three incumbents, we compared the performance of the proposed CR using five or ten energy detectors, with the models presented in the technical literature with one energy detector, that were defined as BaselineSVM and BaselineMP.

Figure 23 and Figure 24 show that the proposed CR is better than the baseline models. It is worth noting that the result remained more reliable even when the CR used the mathematical model with ProbSuccess of at least 0.90. Figure 23 and Figure 24 show a cutoff in BaselineSVM and BaselineMP values above 60%. This occurs because, in the simulation of the technical literature models, there were no attempts to occupy the incumbents’ bandwidth, that is, the ML classifiers always evaluated the bandwidth as occupied above 60%.

## 8. Conclusions

This paper proposed one architecture for a CR that explored opportunities at the 2.5 GHz band in indoor locations that are currently allocated only to mobile cellular networks, generating low efficiency in the use of the spectrum, but not limited to this environment, because, in large countries it is normal to find regions with poor radio frequency cover that can use this framework. For example, rural areas that can improve wireless technology to increase productivity in agriculture and the quality of life of people living in these areas.

This architecture is based on the study of the properties of the LTE signal by means of real data collected with a spectrum analyzer. For this study, we employed both statistical methods and ML. The study revealed that ML models can provide good predictions for when the incumbent’s bandwidth becomes idle based on information from a small number of RBs with only five energy detectors of 10 MHz and ten energy detectors of 20 MHz—10% of the RBs’ bandwidth. This is important because it simplifies and reduces the cost of the CR. Moreover, the ML models are also capable of accurately estimating the channel occupancy rate in loco. We also showed how this last information can be used in a simple statistical model that predicts the duration of the idle periods, based on a specified level of reliability. It is important to note that other ML algorithms can be tested in this framework and that, most importantly, every methodology has been developed using real data, proving that it is possible to increase efficiency and QoS with unused spectrum to its full potential. 

The results show that the proposed CR, in occupancy rates up to 65%, can predict whether the channel is idle with reliability ranging from 91% up to 99%, and also achieve recalls ranging from 63% up to 99%. Likewise, an efficiency above 80% was obtained in the transmission of 5 ms packets in all incumbents. This paper is different from the methodologies presented in the technical literature, that use traffic data generated by mathematical curves and that present difficulties in expressing good results in low occupancy rates where the use of CR becomes even more important, mainly in licensed spectrum bands and in the area of mobile phone communications.

In addition, in our work we present a CR with good results in the metrics of reliability, recall, and efficiency in the transmission of packets. However, CR with more energy detectors can be evaluated in the development of more specific products that do not have energy consumption restrictions.

We present comparisons with the CR model and the technical literature that, despite being able to perform cognition functions without being linked to an incumbent or database, has limitations when implemented in bandwidths such as LTE, which is divided into smaller units (RBs) by network traffic independently. Thus, we proved that the CR architecture proposal with only five or ten energy detectors presents better results in bandwidths with these characteristics. However, better results can be achieved when using our framework in conjunction with the DDPG concept in environments with multiple CRs.

We evaluated the use of our framework in the incumbent’s bandwidth on an individual basis. Therefore, considering each incumbent as a sub-band of the downlink—2620 up to 2690 MHz with five incumbents—of the LTE band, our framework can be combined with a multi-task deep Q-network (DQN) that is utilized to solve the underlying problem, where communication over each sub-band represents a single task. 

In future work, we intend to investigate the cooperation of multiple CRs in the same indoor environment, exchanging information and increasing spectrum efficiency with IoT or other services where the incumbent’s occupancy rate is up to 65% using DDPG and DQN, as well as evaluating the impact on false positives resulting from the spacing between RBs when considering propagation environments with fading.

## Figures and Tables

**Figure 1 sensors-23-04914-f001:**
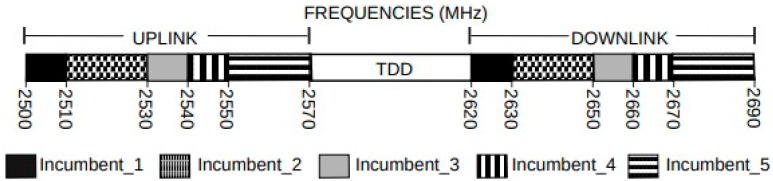
Incumbents and their bands in Rio de Janeiro.

**Figure 2 sensors-23-04914-f002:**
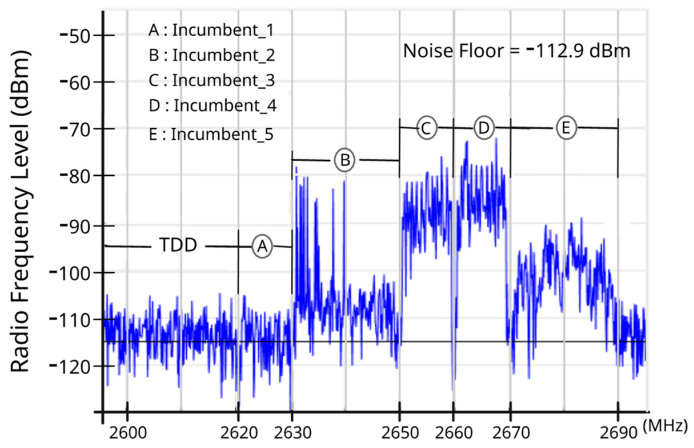
One trace collected during the measurements (1001 points).

**Figure 3 sensors-23-04914-f003:**
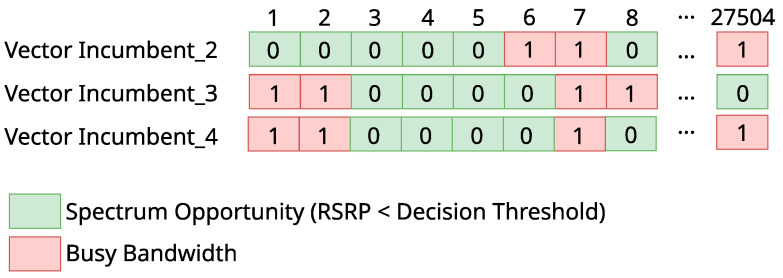
Illustration of spectrum opportunity vectors created from the traces for each incumbent.

**Figure 4 sensors-23-04914-f004:**
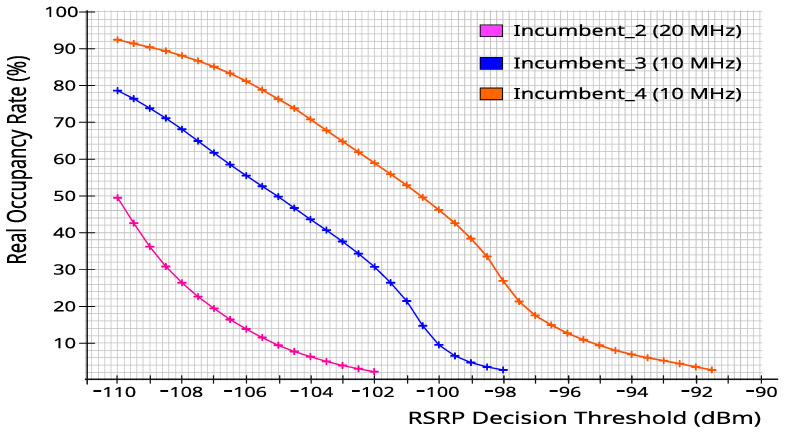
RSRP variation versus real occupancy rate for each incumbent.

**Figure 5 sensors-23-04914-f005:**
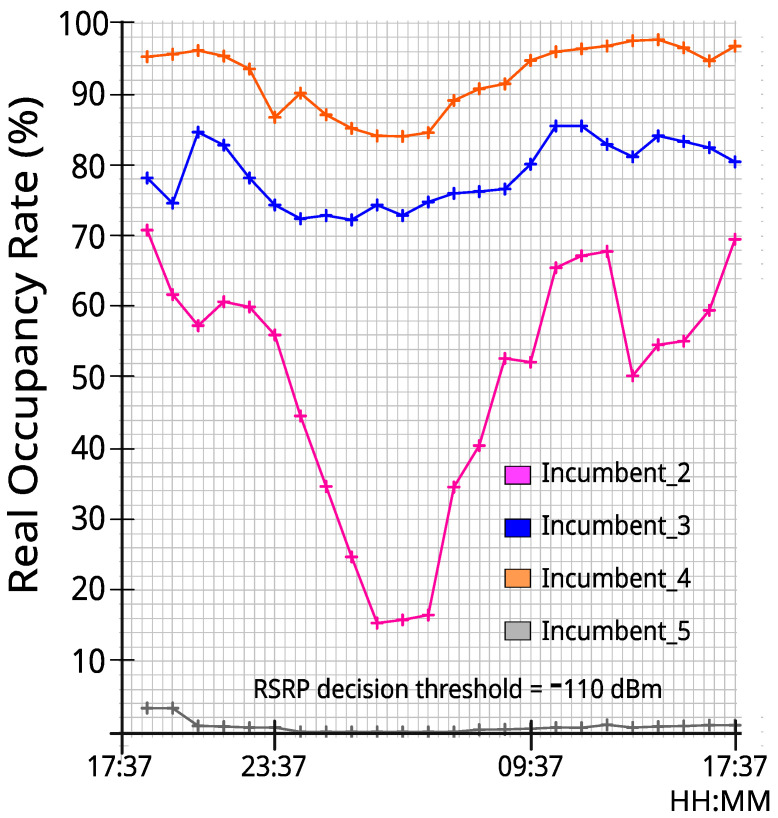
Variation in the occupancy rate for each incumbent during the 24 h period of the measurements.

**Figure 6 sensors-23-04914-f006:**
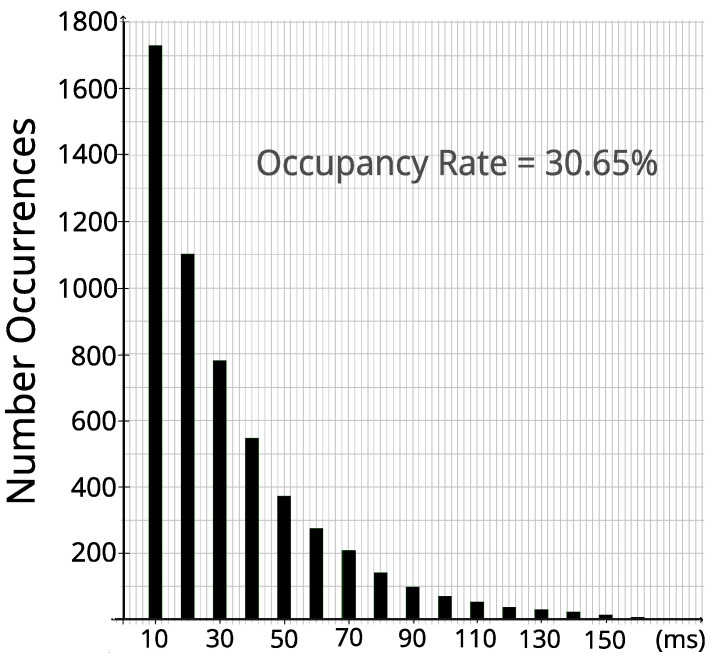
Frequency of occurrence of idle periods of different lengths for the 24 h period of Incumbent_3.

**Figure 7 sensors-23-04914-f007:**
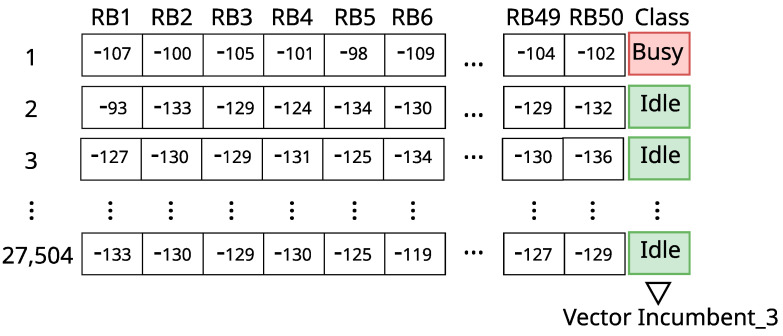
Incumbent_3′s dataset with RSRP threshold = −105 dBm.

**Figure 8 sensors-23-04914-f008:**
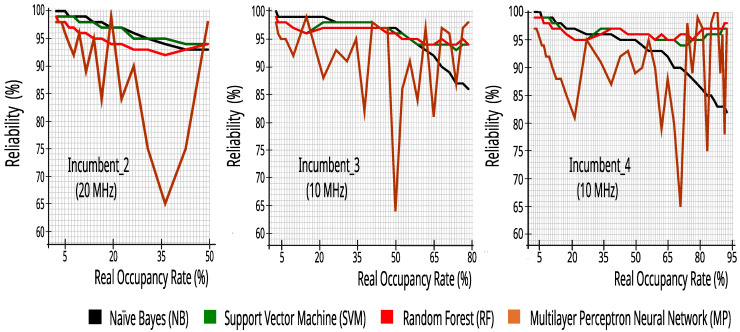
Reliability versus real occupancy rate with 27,504 traces.

**Figure 9 sensors-23-04914-f009:**
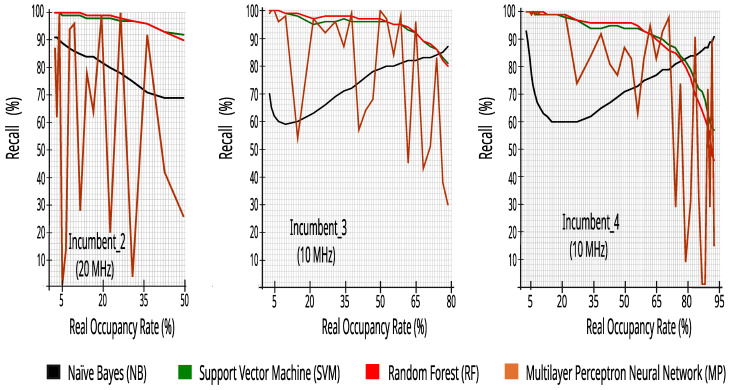
Recall versus real occupancy rate with 27,504 traces.

**Figure 10 sensors-23-04914-f010:**
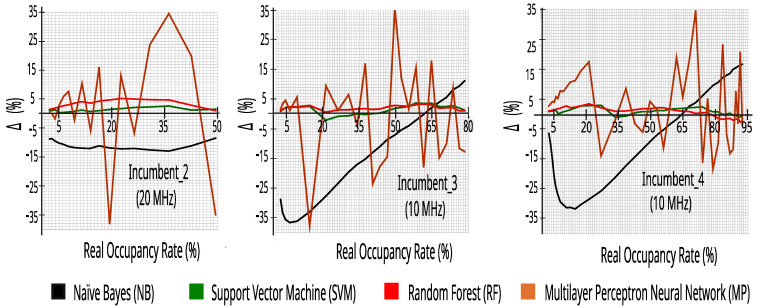
∆ versus real occupancy rate with 27,504 traces.

**Figure 11 sensors-23-04914-f011:**
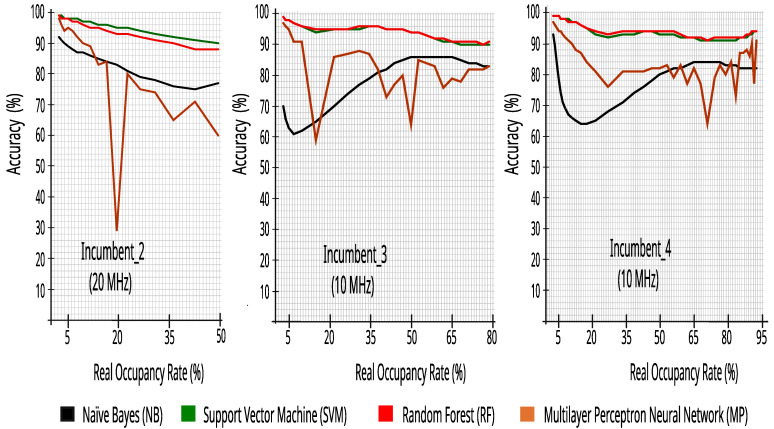
Accuracy versus real occupancy rate with 27,504 traces.

**Figure 12 sensors-23-04914-f012:**
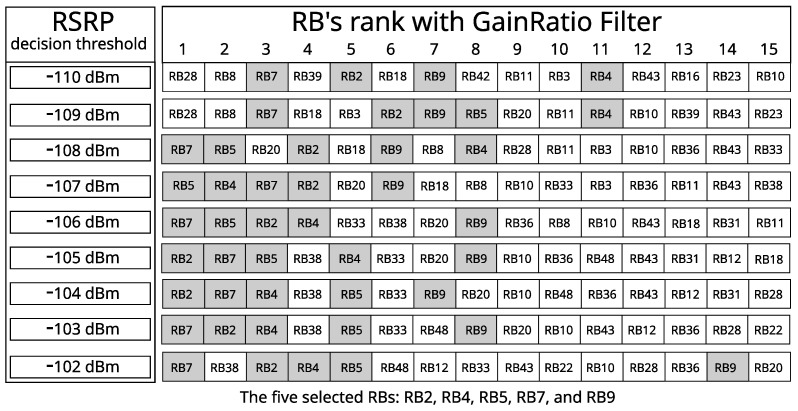
Selection of RBs with GainRatio filter in Incumbent_3.

**Figure 13 sensors-23-04914-f013:**
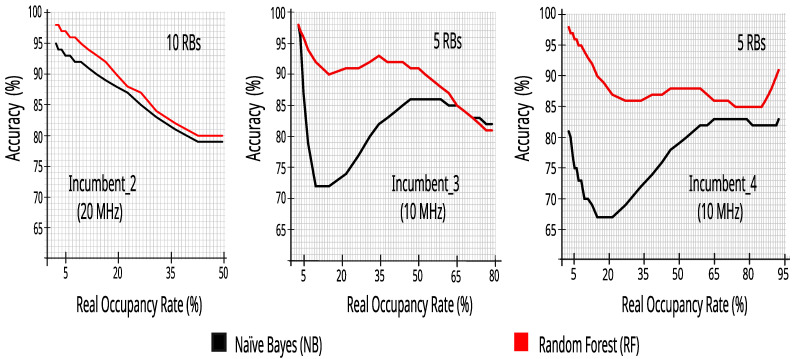
Accuracy versus real occupancy rate with 27,504 traces with 10% of RBs.

**Figure 14 sensors-23-04914-f014:**
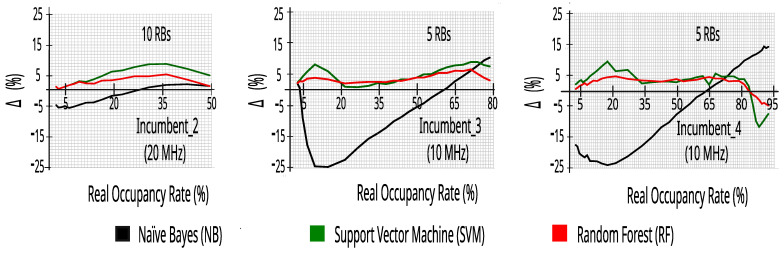
∆ versus real occupancy rate with 27,504 traces with 10% of RBs.

**Figure 15 sensors-23-04914-f015:**
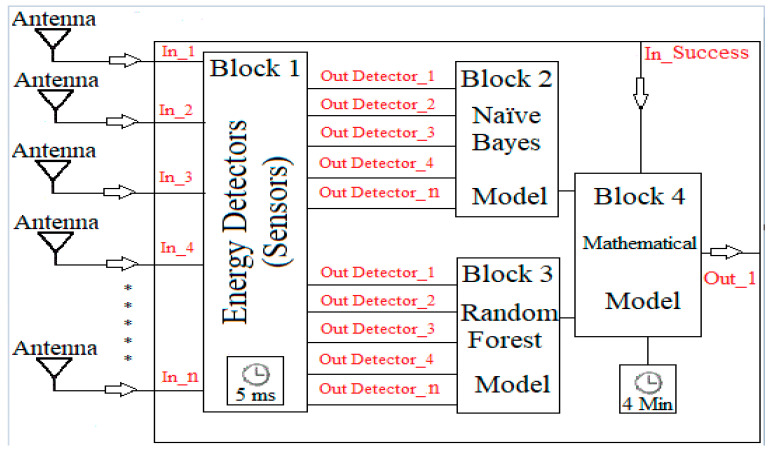
CR architecture proposal.

**Figure 16 sensors-23-04914-f016:**
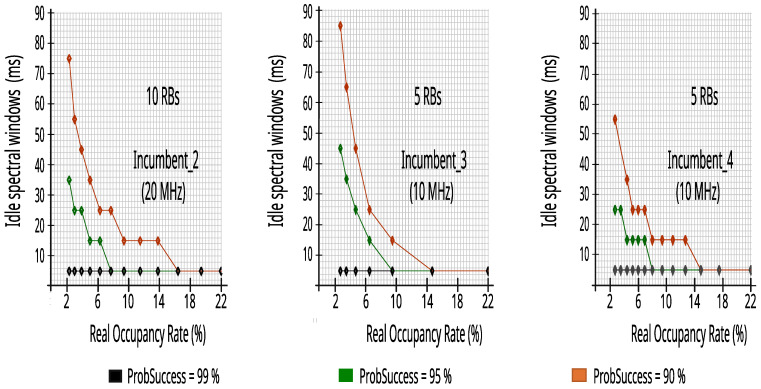
Spectral windows to CR access of LTE bandwidth.

**Figure 17 sensors-23-04914-f017:**
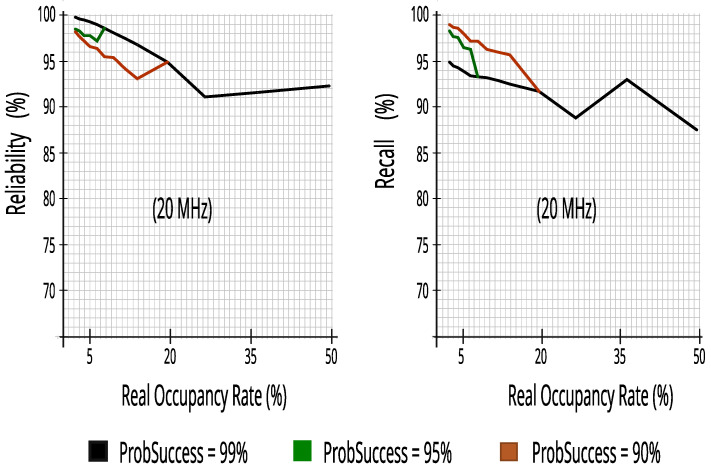
Reliability and recall with ten energy detectors in Incumbent_2.

**Figure 18 sensors-23-04914-f018:**
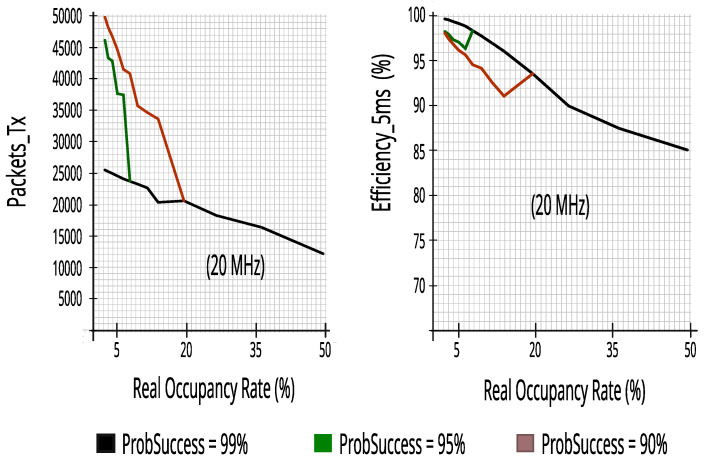
Packets_Tx and efficiency with ten energy detectors in Incumbent_2.

**Figure 19 sensors-23-04914-f019:**
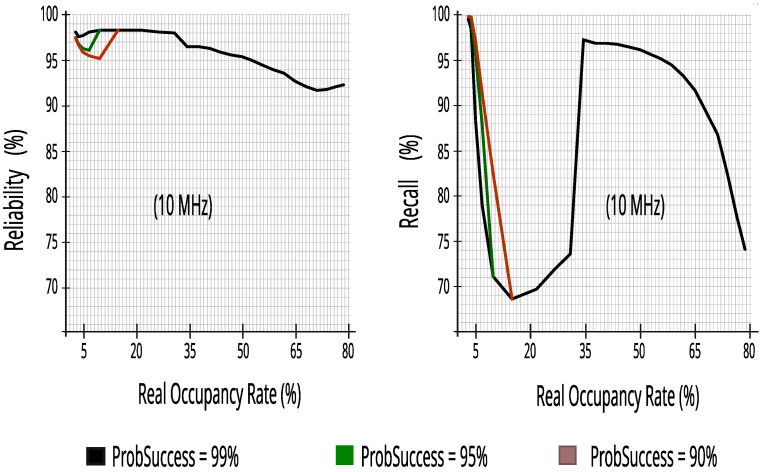
Reliability and recall with five energy detectors in Incumbent_3.

**Figure 20 sensors-23-04914-f020:**
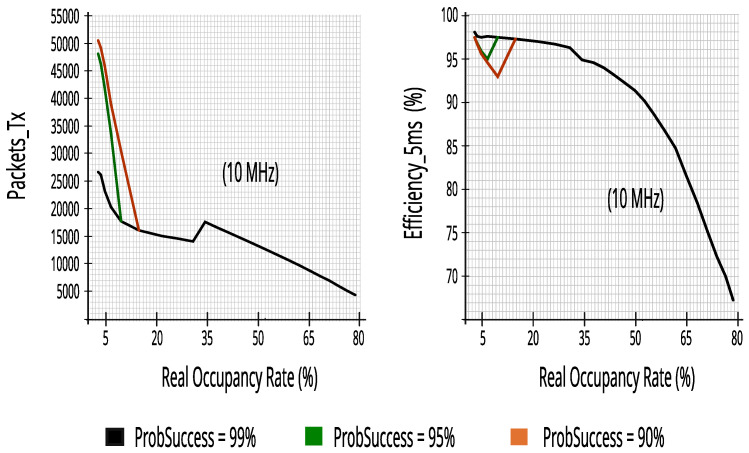
Packets_Tx and efficiency with five energy detectors in Incumbent_3.

**Figure 21 sensors-23-04914-f021:**
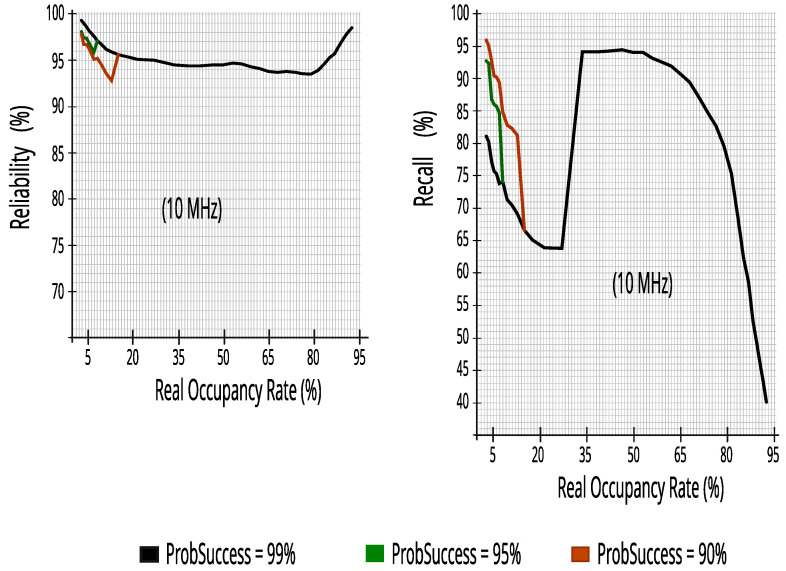
Reliability and recall with five energy detectors in Incumbent_4.

**Figure 22 sensors-23-04914-f022:**
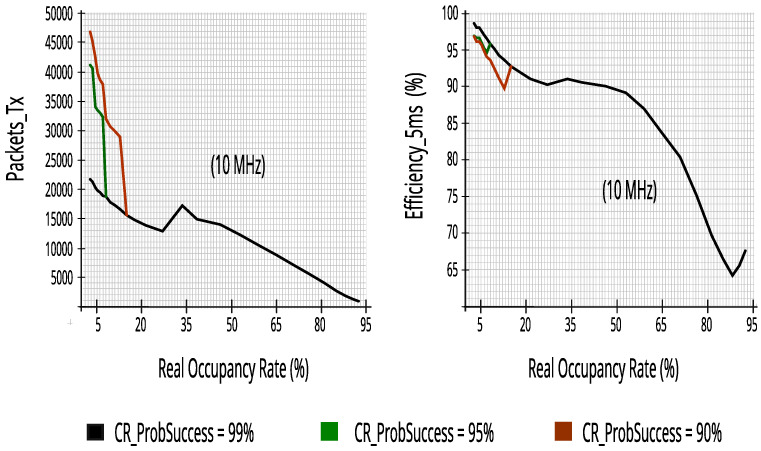
Packets_Tx and efficiency with five energy detectors in Incumbent_4.

**Figure 23 sensors-23-04914-f023:**
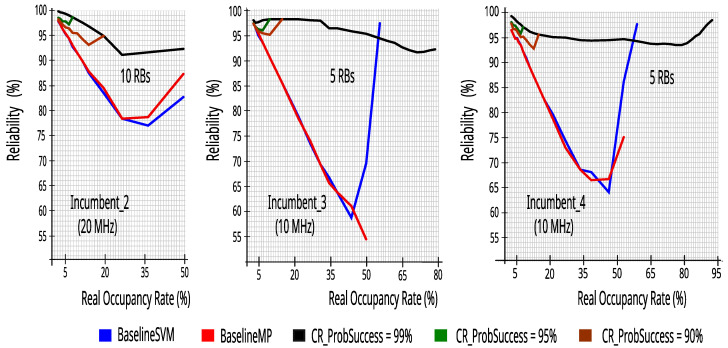
Reliability comparison of proposed CR versus the technical literature.

**Figure 24 sensors-23-04914-f024:**
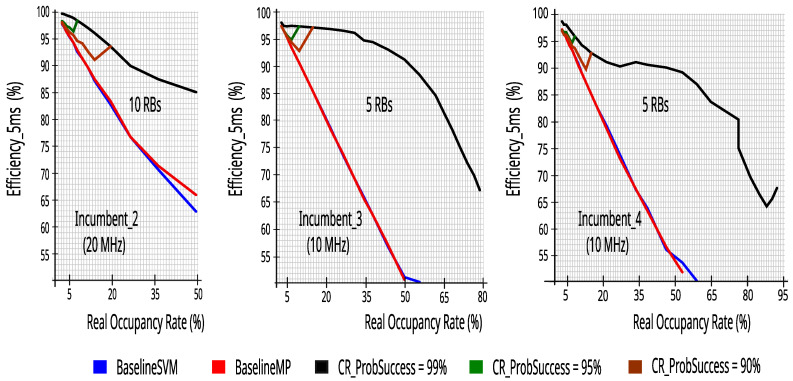
Efficiency_5 ms comparison of proposed CR versus the technical literature.

**Table 1 sensors-23-04914-t001:** Number of traces required to achieve a 95% confidence level at various occupancy rates.

Occupancy Rate (%)	Nº of Required Traces
2	18,832
6.67	5850
10	3900
15	2600
20	1950
30	1300
40	975
50	780
60	650
70	557
80	488
90	433

## Data Availability

The data presented in this study are available on request from the corresponding author.

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
