# Peer review of "Cognitive Radio with Machine Learning to Increase Spectral Efficiency in Indoor Applications on the 2.5 GHz Band"

_sensors, 2023, doi:10.3390/s23104914_

Round 1
Reviewer 1 Report
This paper proposes an architecture for a CR that explore opportunities at the 2.5 GHz band in indoor locations allocated for a mobile cellular network. This paper is easy to follow, and the structure of this paper is clear. Some issues should be addressed as follows.
1. In the abstract, authors use the first three sentences to introduce the motivation and background of this work, which are suggested to be shortened.
2. Authors are suggested to further polish the abstract to highlight the contribution.
3. In the related works, authors should make great efforts to update latest references on cognitive radio networks, deep learning approach, and dynamic spectrum access. The format of references needs to be revised, and a large part of references are published more than five years ago. Recent works about cognitive radio networks, deep learning approach are suggested to be introduced, and I only list a few of them: Spectrum-agile cognitive radios using multi-task transfer deep reinforcement learning, IEEE TWC, 2021; Impacts of sensing energy and data availability on throughput of energy harvesting cognitive radio networks, IEEE TVT, 2023.
4. In the related works, authors need to introduce the content of references in an orderly and logical manner.
5. The machine learning models used in this paper have been studied extensively. Authors need to provide the reasons for using Naïve Bayes, Random Forest, Multilayer Perceptron Neural Network, and Support Vector Machine algorithms.
6. Could the deep reinforcement learning algorithms, such as DDPG, DDQN, be implemented to leverage the use of those opportunities?
7. The readability of figures should be improved, especially Figures 13-14, 16-18.
8. In the simulations, authors only describe the observations from Figures without providing the reasons for these observations, such as Figures 13-14
9. The format of shortening the first line of the paragraph needs to be followed.
This paper is easy to read, and only minor editing of English language is required.
Author Response
Dear Reviewer,
We would like to thank you with all the comments and suggestions that helped us to improve our paper a lot.
Point-by-point answer:
1) We've changed the abstract and would like your feedback on the new version.
2) We think that now is ok. Now we include " The contributions of this work are summarized as: " in the paper.
3) We believe that we revised the formatting items and we also appreciate the article indication, as it helped us with citations and improves our conclusion on the paper.
4) We think that changed the order and include news references to improve the quality of our paper.
5) We used survey references from 2021 and 2019. Reference [17] in our paper is from 2021: 17 A. Upadhye, P. Saravanan, S. S. Chandra, and S. Gurugopinath, “A survey on machine learning algorithms for applications in cognitive radio networks,” Department of Electronics and Communication Engineering, PES University, Bengaluru, India, June 2021. The article we used as a comparison presents models with SVM and MP. Others articles in the literature presented in our paper cite machine learning applications for spectrum sensing with these algorithms.
6) Yes! Thank you for letting us know the references, as we have changed our conclusion and believe that we can use both in combination with our framework to improve spectral efficiency and save energy.
7) Unfortunately, the work had been done in Libreoffice and when we imported it into word, numerous changes occurred. In this way, we revised the entire edition in Word and corrected the text and figures.
8) We think that changes now is ok. Please, could you confirm?
9) We believe that checked all . Please, if you find others send us ?
Feel free to send us more suggestions and comments for improvements.
Best Regards,
Authors

Reviewer 2 Report
This work presents a methodology based on statistical modeling of data collected by a spectrum analyzer and the application of ML to leverage the use of those opportunities by autonomous and decentralized Cognitive Radios, independent of any mobile operator or external database. The proposed design targets using as few narrow-band spectrum sensors as possible in order to reduce the cost of the CRs and sensing time, as well as improving energy efficiency.
Some comments are given below.
-There are many works done on the use of deep learning in cognitive radio networks. Please introduce more related works in this field, like “DDPG-based joint time and energy management in ambient backscatter-assisted hybrid underlay CRNs”.
-What does the symbol nRB in (1) mean?
-Some figures like figure 2 are vague. Also, there are format errors in many figures like Fig. 16-24. The quality of figures should be enhanced.
-Summarize the important conclusions from the measured data.
-about Figure 10, explain why the SVM and RF algorithms have higher accuracy.
The reviewer suggests the further enhancement of writing.
Author Response
Dear Reviewer, We would like to thank you with all the comments and suggestions that helped us to improve our paper a lot.
Point-by-point answer:
1) The suggested work was analyzed and really improve the references of our paper. Therefore, we include it in the references and in the text of the article. The paper is important in the conclusion of our paper too.
2) In the paper we introduce : Equation 1, where n = 50 or 100 for 10 or 20 MHz, respectively, depends on the number of RB on the bandwidth. We believe that now is more easy to reader understand the equation.
3) Unfortunately, the work had been done in Libreoffice and when we imported it into word, numerous changes occurred. In this way, we revised the entire edition in Word and corrected the text and figures. On Figure 2, it tries to present the spectrum sweep in the studied band and the great amplitude of signal variation in each RB, which makes it very difficult for the ML models. We are now trying to inform this in the paper.
4) We tried to improve the text and introduced the contribution part of the work more clearly.
5) We inform about the RF and SVM (Figure 10) and we take the opportunity to highlight the survey in reference (17).
Feel free to send us more suggestions and comments for improvements.
Best regards,
Authors

Reviewer 3 Report
The submitted work carries some novelty, since using ML as a classifier for spectrum bands is not new, however, working in LTE bands based on spectrum campaign looks a novel.
I have some notes and comments on the submitted manuscript
1- The authors are encouraged to improve their way in explaining their methodology, therefore, the readers can easily understanding their work.
2- The related works failed in referring to some important references such as
A-. Mohammed, , et al. "Case study of TV spectrum sensing model based on machine learning techniques." Ain Shams Engineering Journal 13.2 (2022): 101540.
B-Corral-De-Witt, Danilo, et al. "An accurate probabilistic model for tvws identification." Applied Sciences 9.20 (2019): 4232.
C- Malik, Tauqeer Safdar, et al. "RL-IoT: Reinforcement learning-based routing approach for cognitive radio-enabled IoT communications." IEEE Internet of Things Journal 10.2 (2022): 1836-1847.
Authors are encouraged to refer to above mentioned references.
3- It is recommended to rewrite the abstract to show what the author did and their achievements.
4- The presentation quality of the manuscript should be enhanced,
A- I mean the font and style of the equations should be the same.
B- The size of figures should be the same.
C- If you look at the figure 8-11, you can easily notice that the y-axis label overlaps the figure and the legend.
D- Section references needs some improvement
It is okay
Author Response
Dear Reviewer,
We would like to thank you with all the comments and suggestions that helped us to improve our paper a lot.
Point-by-point answer:
1) We tried to improve several points of the article and even made available in the references the number (43) that allows readers to better understand how to adjust the spectrum analyzer and collect samples within an LTE spectrum.
2) The suggested works were analyzed and really improve the references of our paper. Therefore, we include all of them in the references and in the text of the article.
3) The abstract was rewritten with suggestions from all reviewers.
4) The entire article was revised, where we tried to improve the quality of all the points questioned.
Feel free to send us more suggestions and comments for improvements.
Best regards,
Authors

Round 2
Reviewer 1 Report
Authors have made great efforts to address the issues of the first-round review. Some issues still need to be addressed as follows.
1. In the introduction, authors need to improve the logic of introducing related works. It is not appropriate to introduce the contributions of one reference per paragraph. Besides, authors are suggested to adopt the past tense throughout the introduction of related works.
2. In the related works, authors should make great efforts to update latest references on cognitive radio networks, deep learning approach, and dynamic spectrum access. The format of references needs to be revised. Recent works about cognitive radio networks, deep learning approach are suggested to be introduced such as Impacts of sensing energy and data availability on throughput of energy harvesting cognitive radio networks, IEEE TVT, 2023.
3. In the last paragraph of Section 2, authors need to emphasis the motivation and contributions in a clear way, rather than introducing the contents of this work such as “We also tackle practical implementation issues, particularly regarding how to decrease the number of energy detectors required for the CR in 10MHz or 20 MHz Incumbent’s bandwidth”.
4. The punctuations in the formulas should be added.
5. The figures need to be embedded in the paper in eps format so that it can be displayed more clearly. Besides, some figures should be revised for readability, such as figures 8-9.
6. The traditional value range of the RSRP threshold needs to be provided and justified.
7. The formulas should be referred according to the format, such as “one can solve Equation 3 for TMAX which yields:” should be revised as “TMAX could be solved by (3) as”.
8. Authors are highly suggested to check the writing and grammar issues throughout this manuscript, and improve the writing and logic of this manuscript to show the technical depth and contributions.
Authors are highly suggested to check the writing and grammar issues throughout this manuscript, and improve the writing and logic of this manuscript to show the technical depth and contributions.
Author Response
Dear Reviewer,
We would like to thank you with all the comments and suggestions that helped us to improve our paper a lot.
Sorry, but in the last revision we forgot to inserte " Impacts of sensing energy and data availability on throughput of energy harvesting cognitive radio networks" . Thanks to remember us about so important reference. Now you can see it in our paper [19] - the publication was 2023!
We changed the others itens and put ( color red) in the new version how can you check it in our paper.
Best Regards,
Authors

Reviewer 2 Report
My comments have been addressed.
Author Response
Thanks a lot !